# Biliary Atresia: Clinical Phenotypes and Aetiological Heterogeneity

**DOI:** 10.3390/jcm10235675

**Published:** 2021-12-01

**Authors:** Mark Davenport, Ancuta Muntean, Nedim Hadzic

**Affiliations:** 1Department of Pediatric Surgery, Kings College Hospital, London SE5 9RS, UK; ancuta.muntean@nhs.net; 2Department of Paediatric Hepatology, Kings College Hospital, London SE5 9RS, UK; nedim.hadzic@kcl.ac.uk

**Keywords:** biliary atresia, Kasai operation, liver transplant, etiology, adjuvant therapy

## Abstract

Biliary atresia (BA) is an obliterative condition of the biliary tract that presents with persistent jaundice and pale stools typically in the first few weeks of life. While this phenotypic signature may be broadly similar by the time of presentation, it is likely that this is only the final common pathway with a number of possible preceding causative factors and disparate pathogenic mechanisms—i.e., aetiological heterogeneity. Certainly, there are distinguishable variants which suggest a higher degree of aetiological homogeneity such as the syndromic variants of biliary atresia splenic malformation or cat-eye syndrome, which implicate an early developmental mechanism. In others, the presence of synchronous viral infection also make this plausible as an aetiological agent though it is likely that disease onset is from the perinatal period. In the majority of cases, currently termed isolated BA, there are still too few clues as to aetiology or indeed pathogenesis.

## 1. Introduction

Biliary atresia (BA) is an obliterative condition of the biliary tract that presents with persistent jaundice, pale stools and dark urine in the first weeks of life and, if left untreated, ultimately leads to cirrhosis and end-stage liver failure (Figure 1). Beyond this unchallenged statement, much of the rest are observational facts and hypothetical speculation. This is certainly the case for its aetiology if not its post-natal pathogenesis [1]. The aim of this chapter is to review the spectrum of BA as it presents to the clinician reinforcing this concept of aetiological heterogeneity as a principle feature of the disease itself.

### Geographical Variation and Incidence

The incidence of BA is markedly variable across the world, ranging from about 1:5–10,000 live births in Taiwan [2] and Japan [3] and presumably in China to about 1:15,000–19,000 in Europe [4,5]. The incidence in North America tends to parallel the latter estimates, with the most recent data based on US billing returns at 1:22,000 [6] and 1:19,000 based on Canadian Registry data [7]. Interestingly, both these later studies have suggested that its incidence has increased over the past 20 years, although the explanation is far from clear and has certainly not been suggested elsewhere. The incidence in other parts of the world such as the Indian sub-continent, South America and Africa is less clear in the absence of national studies. Aetiological heterogeneity is one obvious explanation of such variation with the proportion of different variants changing with the local environment or some genetic predisposition.

Some national studies have looked at racial composition for evidence of variation. Evans et al. reported the national New Zealand series quoting an incidence of 1 in 5285 in those of Māori ancestry compared to about 1 in 16,000 live-births for those of European ancestry [8]. Our own experience in England and Wales supports a significant ethnic variation so there is a significant variation by health region from 1 in 14,000 in London the region with the highest non-white ethnicity (40.4%) to 1 in 22,700 for North-West England with one of the lowest non-white ethnicity proportions (14.6%) (Figure 2). 

Whether there is some kind of genetic predisposition is not known. Most of the work on this subject has emerged from varying parts of the world, but notably China with initial identification of *ADD3* and *XPNPEP1* mutations in a Han Chinese population [9] and more recently a spectrum of biallelic deleterious variants in liver-expressed “ciliary genes” [10]. However, the degree of risk seems small, as the latter study implies only a two-fold increase in risk compared to normal.

## 2. Aetiological Heterogeneity

From a clinical perspective, we can separate BA into distinct categories featuring common characteristics (Table 1).

### 2.1. Syndromic Biliary Atresia

We recognise two syndromes where BA is a key feature (Table 1). The commonest of these is the Biliary Atresia Splenic Malformation (BASM) syndrome, which accounts for about 10–15% of European and American series [11,12], but is distinctly rare in Chinese and Japanese series (<2%) [13,14]. For instance, Zhan et al. reviewed 851 cases from five centres in mainland China [13]. There were only two (i.e., 0.23%) with situs inversus and four (i.e., 0.5%) with polysplenia—both hallmarks of BASM. In Japan, there is a reported incidence of about 4% from the Sendai series [14]. By comparison, we have an incidence of 14% in our national registry of infants from England and Wales (1999–2000) (unpublished observation).

The phenotype of BASM is unmistakable and can be characterised by a host of visceral anomalies. The splenic malformation is typically polysplenia, but sometimes can be asplenia or double spleen [11,15]. The other obvious and evident visceral anomalies are: situs inversus, present in about 30–40% of patients with and without malrotation; preduodenal portal vein and a complete absence of the intrahepatic vena cava. Cardiac anomalies are apparent in about half the cases overall, but there is lack of a consistent cardiac phenotype. Indeed, when we reviewed our experience of biliary atresia-associated cardiac anomalies (*n* = 37), features indicative of BASM were present in 48% [16].

Therefore, how might we explain BASM? The biliary system develops in two distinct phases. Firstly, the extrahepatic bile duct arises as an outpouching of the foregut at Carnegie embryo stage 11 (4th week) and is essentially complete at about Carnegie embryo stage 17 (6th week) with a patent common duct and gallbladder, all in intimate contact with the developing, predominantly haematopoietic, liver. The other constituents within the liver at this stage are the hepatoblasts, which then differentiate into hepatocytes from around 49 days (7th week), and biliary epithelial cells (now expressing SOX9 and CK19), which by a process of selection and deletion around the ingrowing portal venous network initially form the ductal plate and then a more mature tubularising biliary network. Actual contact and interlacing with the extrahepatic duct occur at or around 12 weeks gestational age, just in time to transport newly formed bile from the hepatocytes into the gallbladder and duodenum.

As this biliary timeline is shared by the determination of visceral situs, spleen formation, evolution of the portal venous and caval venous system anomalies it is not too much to speculate that BASM is also an embryonic defect. The actual mechanism is still obscure though, and a genetic mutation has long been sought to explain BASM. Previous case reports suggested mutations in *CFC1*, *NODAL*, *FOXA2* and *ZIC3*, e.g., [17]. However, the first systematic search in a population was only recently carried out by Berauer et al. [18] using whole exome screening of 67 infants with the “BASM” phenotype and in 58 including their parents as a trio analysis. This being a good example of the benefits of a large multicentre biobank—the ChiLDReN network. It is worth noting, however, that only 60% of this group actually had a splenic malformation, with four being counted despite only having isolated renal anomalies—something not a part of the original definition [15]. Five children with “BASM” had rare biallelic variants in the gene, polycystin 1-like 1 (*PKD1L1*) found on chromosome 7. There is also a biologically plausible pathogenic mechanism, at least in the mouse, as *Pkd1l1* heterodimerizes with *Pkd2l1* in primary cilia, to form a transmembrane ciliary calcium channel that ultimately influences downstream Hedgehog signalling. Such heterodimers are required to establish normal left–right asymmetry and are an obvious fit for the genesis of situs inversus.

If not genetic then what? Well, the original BASM series also identified a link with maternal diabetes and possibly other first trimester ‘insults’, which influence the embryonic environment though these still remain poorly defined [10,15]. The National Birth Defects Prevention study from the USA also recently identified a significant association between first trimester use of bronchodilators and anti-inflammatory medication and subsequent biliary atresia in their offspring. Whether the cases identified were syndromic in any way is not known [19].

The Cat-eye syndrome (CES) or at least, aneuploidy of chromosome 22, is less well described, but its association with BA seems clear [20]. We originally described five infants (4 with BA) from a combined London and Paris series with a range of genetic anomalies including classical Cat-eye syndrome, partial duplication of chromosome 22 (supernumerary der(22) syndrome), and a mosaic for trisomy 22. Clinically, these infants typically have coloboma, cardiac anomalies and anorectal malformations. Some have even had neonatal surgical procedures before the biliary association was recognised.

There are also other defined syndromes which may have BA as a component though are much less frequently reported. Some examples from our National BA series of over 800 infants were Kabuki syndrome [21] (characteristic facial features, skeletal anomalies and mild developmental delay), Zimmermann–Laband syndrome (cranio-facial and oral abnormalities including gingival fibromatosis), Kartagener syndrome (ciliary motility pathology causing situs inversus, chronic sinusitis and bronchiectasis) and perhaps the much more common Hirschsprung disease (Table 2). In some of these, there were also overlapping features with BASM—both of the infants with Kartagener syndrome for instance. 

There also appears to be a non-random association (i.e., more than would be expected by chance) with some other otherwise isolated anomalies such as oesophageal atresia, duodenal atresia, jejunal atresia, cleft palate, etc. (Table 1). Again, some showed an obvious overlap with BASM (e.g., duodenal atresia with 10/13 cases), while others did not (e.g., oesophageal atresia with 1/8 cases).

We have recently characterised another sub-group defined purely by its association with cardiac anomalies (cardiac-associated biliary atresia or CABA) [16]. While we do not claim that this has a uniform pathogenesis, it is clearly a high-risk subgroup and one of the main contributors for actual mortality in BA overall. As an aside, we have recommended a ‘heart-first’ strategy with restorative cardiac surgical physiology preceding KPE if possible, to improve both liver outcome and overall survival.

### 2.2. Cystic Biliary Atresia

Cystic changes, usually containing mucus, but sometimes bile, can also be found at the level of the otherwise obliterated extrahepatic biliary tree [22,23,24]. This is cystic BA (CBA), and care needs to be taken to avoid being misdiagnosed as a congenital choledochal cyst. Both may be antenatally detected on the maternal ultrasound, usually around the time of the feta anomaly scan at 18–20 weeks of gestation, though the former changes are usually consistently smaller [23]. This distinction is important, as they have a different clinical course. All those with CBA will remain jaundiced with pale stool, while some neonates with cystic choledochal malformations may actually clear the jaundice and have normally pigmented stool. Timely operative cholangiography is the key investigation in a jaundiced infant with a postnatally confirmed subhepatic cyst. In CBA, this may show a connection with the intrahepatic ducts or ductules and it is usually tenuous and clearly abnormal, often being described as “cloud-like” [22]. 

This, at least, implies an onset beyond 12 weeks (to allow bile to come into the common duct), which is completely developed by 16–18 weeks, the earliest point at which antenatal detection might be made. This phase is co-incident with the arterialisation of the liver, and one could speculate that there may have been some ischaemic event affecting the distal extrahepatic duct with consequential proximal dilatation. 

Early studies showed that it was possible to reproduce the key features of CBA in experimental models: by ligation of the common bile duct in foetal lambs at about 80 days of gestation, and by ligation of the hepatic artery in foetal rabbits [25,26,27]. Not only can this produce cystic extrahepatic change, but also in a proportion impairment of the intrahepatic bile ducts as well [27]. More recently, a group from Porto Allegre, Brazil have looked at the possible role of ischaemia in reproducing the cholangiopathy of (isolated, not necessarily CBA) BA by hepatic arterial morphometry and expression of angiogenesis mediators. BA specifically seems to be characterised by an increase in arterial medial layer thickness at the time of portoenterostomy compared to controls and becoming progressive in those requiring liver transplant [28]. Furthermore, gene expression of hypoxia-inducible factors (HIF), HIF1a and HIF2a were increased in BA cases, while vascular endothelial growth factors (VEGFA) (VEGFR1 and VEGFR2) were decreased suggesting reduced angiogenesis [29]. Whether these observations are indicative of an aetiological factor or in some way secondary to the inevitable changes wrought by fibrogenesis is not known. 

Most CBA cases, even those with bile-filled cysts, should still come to a radical resection and wide portoenterostomy rather than attempt to preserve any part of the cyst. Post-operatively, these infants have >75% chance of clearance of jaundice and native liver preservation, though their prognosis does appear to have a marked relationship with age at surgery [30]. Certainly by comparison with the other variants, these children have a better long-term prognosis [31], though our recent review of 20 year follow-up in the national BA registry showed that a significant proportion of these seemed to decompensate requiring liver replacement during their transition to adulthood (unpublished observation). 

### 2.3. Cytomegalovirus-Associated BA [See Also Fischler et al. CMV and BA in Same Issue]

In 1974, the American paediatrician, Benjamin Landing, proposed that a perinatal viral infection might be one of the origins of BA [32]. Nevertheless, he was not too specific in this pronouncement, and also proposed the same for choledochal cysts and neonatal hepatis as well. Several candidate viruses have been suggested over the years, with the original being REO-virus Type 3 both by serological [33] and PCR studies [34], though this has been disputed by more recent Japanese evidence involving a much bigger numbers of patients [35] and a review of published studies [36]. Indeed, the relevance of any viral identification was questioned by Rauschenfels et al. from Hannover who, using multiple viral PCR primers, identified viral genetic material in a significant proportion, but felt this was more likely to be a secondary phenomenon [37].

Of all the candidate hepatotropic viruses, perinatal cytomegalovirus (CMV) infection, a double-stranded DNA virus from the Herpesviridae family, has received most attention. The relationship was first suggested by Bjorn Fischler and a Swedish group in 1998 observing a high proportion of their cohort of BA with signs of CMV infection [38].

It is clear that this virus can be detected in a variable proportion of cases of BA, but this too has shown marked global variation. Therefore, using CMV IgM antibodies as a marker of infection up to 30% of Chinese BA series have been positive [39] compared to about 10% in a UK series [40]. Furthermore, in a series from Denver, CO, 55% of BA cases were shown to have CMV-specific T cell responses at the time of surgery also suggesting early exposure [41].

What is not known is the timing of exposure of the virus in these infants. We have no additional data of their CMV status—i.e., antenatal maternal CMV serology or whether for instance their neonatal screening blood spot tests were also IgM positive.

In our experience, CMV IgM positive infants do have distinct clinical and histological features compared to those with IBA, such as: an older age at diagnosis; larger ultrasound-measured spleen sizes and a greater degree of histological liver inflammation and fibrosis, even when that is corrected for post-natal age [40]. Interestingly it can also be shown that they have a Th1 predominant mononuclear cell infiltrate, again compared to those with IBA and even the syndromic BA infants [42]. 

At least at Kings College Hospital, these infants, by comparison to CMV IgM negative controls, have a poorer outcome with a reduced clearance of jaundice and native liver survival together with a demonstrable increase in actual mortality [40]. A more recent systematic analysis of published evidence also seemed to support CMV as a negative prognostic factor [43].

The actual mechanism of the cholangiopathy in such infants is intriguing and may not be simple direct cholangiocyte damage by the virus. Rather, it is believed to be a more subtle auto-immune process with the virus triggering self-damage, allowing perpetuation of a pro-inflammatory immune response driven by macrophages, NK cells, Th1 and Th17 cells and unrestrained by a postulated deficit in regulatory T cells. The details of this aspect are outside the remit of this article, but the concepts are illustrated by Figure 3, based on Kilgore and Mack [44] and Ortiz-Perez et al. [45].

### 2.4. Isolated BA 

The term “isolated” BA is used when there are no other defining characteristics, and unfortunately, this is the largest clinical grouping with no real hint as to its aetiological mechanism. Nevertheless, these still might include genetic [9,10], developmental [10], ischaemic [28], environmental [46] and other viral causes [34,35,38,40,44,45]. 

There may well be some kind of non-Mendelian genetic predisposition to the development of isolated BA, although the scale of this is completely unknown. The relationship between *ADD3* and *XPNPEP1* mutations has been mentioned before [9], though more recent studies involving genome-wide association studies (GWAS) in various discrete populations have also implicated *GPC-1* [47]. The latter is expressed in biliary epithelial cells and is involved in inflammatory mediators making a contribution to biliary pathology plausible. Other recent studies have used whole exome sequencing (WES) in small cohorts of clinically diagnosed BA cases throwing up candidate genes that have included those involved in the ABC superfamily, and the Notch signalling pathway (JAG1) [48]. Finally, there may be mutations which can modify the response to treatment (i.e., Kasai portoenterostomy). Mezina and Karpen [49] reported a greater frequency of variants (e.g., p.A934T) in the gene encoding the phospholipid floppase, *ABCB4* in those who required early liver transplant compared to those with a good outcome. Whether these are really significant either as predisposing or modifying elements remains to be seen. Clearly, there may be overlap, at least genetic, with a variety of other, more distinct, neonatal cholestasis syndromes such as Alagille’s syndrome and PFIC.

## 3. Pathological Classification

At surgery, and essentially unrelated to the foregoing descriptions, the pathological type of BA is defined by the most proximal level of biliary obstruction. Type 1 BA (5–10%) is where obstruction to bile flow is at the level of common bile duct, and typically bile is found in the gallbladder. The proximal biliary tract is often cystic in these. In Type 2 BA, the obstruction is at the level of the common hepatic duct and dissection within the porta hepatis will show two distinct, albeit thick-walled and abnormal, hepatic ducts. This is exceedingly rare in most series (1–2%). By contrast, Type 3 BA, is by far the most common (>90%) with its obstruction level high within the porta hepatis and in these there are no visible macroscopic ductules present—the transected porta presenting a fairly uniform bland appearance. 

BASM is somewhat of an exception to the foregoing, as there is a characteristic appearance of the remnant extrahepatic bile duct. There is usually a small solid gallbladder and absent common bile duct, with often quite defined proximal segmental branching in and around the abnormal portal vasculature. 

## 4. Timing of Disease Onset

This is an important principle in trying to identify when BA actually occurs. We have already made the case for intrauterine onset in BASM, and the other syndromic and non-syndromic associations; and for cystic biliary atresia, but for the largest grouping of isolated and indeed viral-association BA we really have little actual evidence. Is isolated BA truly a congenital anomaly present at birth or is it acquired somewhat later? The pendulum has swung on this over the years, as initially it was felt to be mostly perinatal in onset with a period of normal pigmented stools and later onset of jaundice. 

Still there are two important areas of study which might tilt the balance of opinion. The first arose from obstetric observations initially made by a French group [50]. γ-glutamyl transpeptidase (GGT) is specifically secreted by biliary epithelial cells and is normally found in high levels in amniotic fluid during the second trimester due to passage of bile into the foetal intestine. Its level normally tails off during the third trimester as the foetal anal sphincter closes. In cases that were later shown to be BA, this second trimester rise does not happen suggesting that the bile flow had already ceased. A more recent series of clinical experience with amniotic fluid GGT measurement has been reported by an Israeli group and combined with non-visualisation of the foetal gallbladder [51,52]. Perhaps surprisingly, this latter observation is not usually associated with BA. However, amniotic fluid GGT was measured in 32 cases with the non-visualisation in this series and found to be low in five. In three of these, a postnatal diagnosis of BA was made. 

The second key evidential observation for the isolated form comes from the work of Sanjiv Harpavat’s group in Houston, Texas [53]. Initially, they retrospectively screened fractionated bilirubin levels in BA infants and showed that their direct conjugated bilirubin was abnormally elevated in all 34 BA patients at day 1 and 2 of life [54]. Subsequent studies from the same group have confirmed this key observation [55] and also raised considerable expectations for neonatal screening for BA at least in North America.

## 5. Conclusions

BA is still a fascinating enigma of a disease with much to unravel as far as its clinical manifestation. It seems unlikely we will uncover a universal hypothesis to explain its aetiology sometime soon and appropriate caution should accompany those laboratory findings which assume insight, particularly when based on experimental animal models without parallel in the natural world. It seems far more likely that the infants that we see both in the operating room and the clinic have arrived there by a multiplicity of possible pathways.

## Figures and Tables

**Figure 1 jcm-10-05675-f001:**
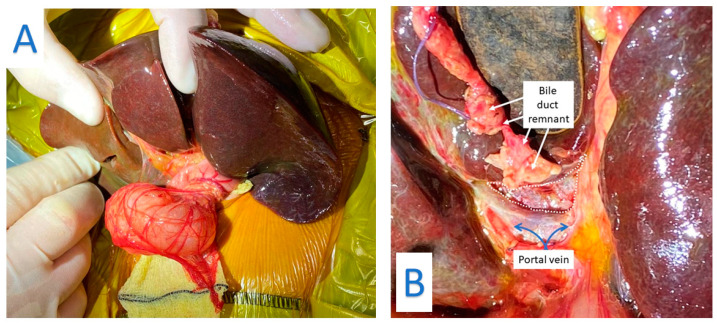
(**A**) Biliary atresia: Liver is mobilized and exteriorized to expose the porta hepatis. (**B**): Dissection of the Porta Hepatis. The bile duct remnant has been transected and is lying on segment 4 of the retracted liver. The white dotted area outlines the extent of the porta hepatis which will then be anastomosed to the Roux jejunal loop.

**Figure 2 jcm-10-05675-f002:**
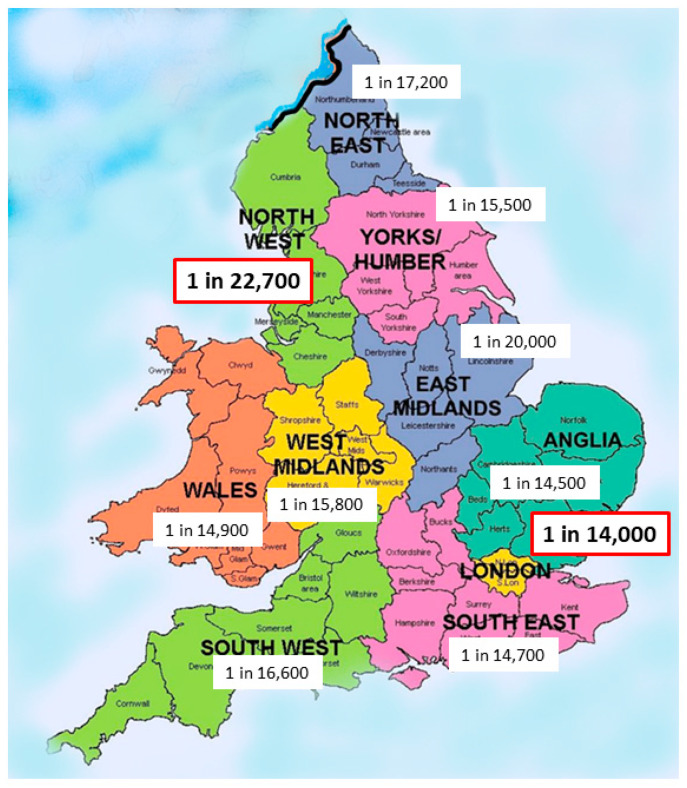
Variation in prevalence of Biliary Atresia in England and Wales (*n* = 713, 1999–2015).

**Figure 3 jcm-10-05675-f003:**
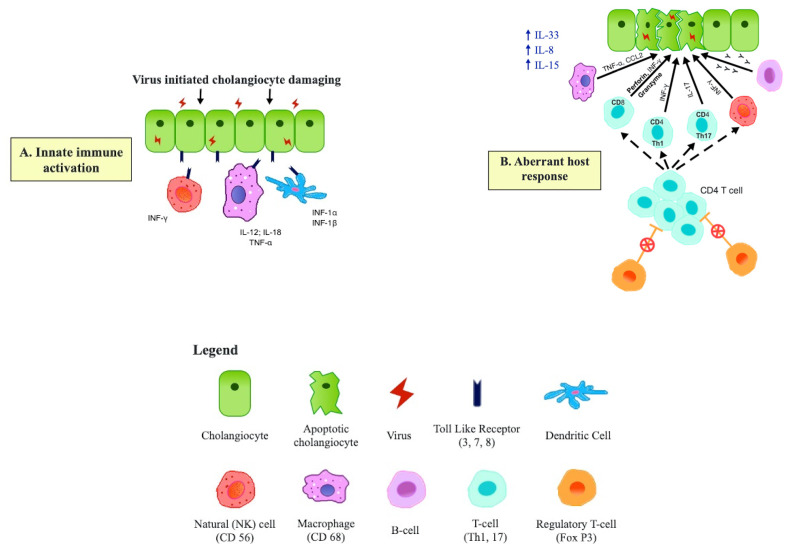
Suggested immunopathogenesis of Biliary Atresia. (**A**) Transient virus infection of cholangiocytes results in upregulation of Toll-Like Receptors (TLR) and a broad-based activation of the innate immune system involving macrophages, dendritic cells and NK cells. (**B**) Adaptive T cell proliferation (Th2 and Th17 predominant), supported by B cells and activated macro-phages cause cholangiocyte damage, possibly unrestrained by the absence of Tregs. Dissection of the Porta Hepatis. The bile duct remnant has been transected and is lying on segment 4 of the retracted liver. The white dotted area outlines the extent of the porta hepatis which will then be anastomosed to the Roux jejunal loop.

**Table 1 jcm-10-05675-t001:** Etiological heterogeneity—clinical categories of Biliary Atresia.

Category		Associated Clinical Features
**Isolated BA (70–80%)**		
**Syndromic BA (10–15%)**	**BASM**	Polysplenia, asplenia, situs inversus, pre-duodenal portal vein, absence of IVC, CHD, malrotation.
	**Cat-eye syndrome**	Coloboma, ano-rectal atresia, CHD et al.
	**“Non-syndromic”**	e.g., Esophageal atresia, jejunal atresia, cleft palate et al.
**Cystic BA (5–10%)**		Antenatal or postnatal detected cyst at porta hepatis.
**CMV IgM + ve BA (~10%)**		Defined by CMV IgM + ve antibodies. ↑ age at KPE, ↑AST ↑spleen size Th1 predominant mononuclear infiltrate in liver

CHD—congenital heart disease, IVC—inferior vena cava, AST—aspartate aminotransferase, CMV—cytomegalovirus, KPE—portoenterostomy.

**Table 2 jcm-10-05675-t002:** Associated anomalies and structural biliary anomalies in the National England and Wales Biliary Atresia Registry (January 1999–December 2019) *n* = 867.

Anomaly	Total N (%)	Notes and Overlap
Recognised Syndromic Association
BASM	122 (14.1%)	
Cat-Eye/Emanuel syndrome	7 (0.8%)	
Possible Syndromic Association
Kabuki syndromic	3	
Kartagener’s syndrome	2	BASM (*n* = 2)
Hirschsprung’s disease	2	Cat-eye syndrome (*n* = 1)
Zimmermann-Laband syndrome	1	
Gastrointestinal Anomalies
Duodenal atresia	13 (1.5%)	BASM (*n* = 10)
Ano-rectal anomalies	5	BASM (*n* = 1)
Oesophageal atresia	8 (1%)	BASM (*n* = 1)
Jejunal/ileal atresia	4	BASM (*n* = 3)
Pyloric stenosis	1	Ch6p deletion
Other Anomalies
Cardiac anomalies (isolated)	6	Ring Chromosome18 (*n* = 1)
Cleft lip/palate	6	
Isolated Anomalies
Exomphalos	1	
Gastroschisis	1	
Spina bifida	1	
Choanal atresia	1

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
