# Peer review of "Biliary Atresia: Clinical Phenotypes and Aetiological Heterogeneity"

_jcm, 2021, doi:10.3390/jcm10235675_

Round 1

Reviewer 1 Report

The Authors proposed a very good paper about classification, pathogenesis, and clinical implications on BA.

However, the actual form of the paper resembles the chapter. Thus, it should be re-typed into manuscript/review paper. Methods were not described - it was a literature review enriched with their own observations.

Author Response

Thanks for your observations... It is a literature review, with our observations.  That was the format requested by the editor. As such it does not require a "methods" section as such. Certainly I do not propose to turn it into a "systematic review" with all that entails.

Reviewer 2 Report

The manuscript is a comprehensive review on pathogenesis of biliary atresia with a focus on the concept of common inflammatory pathway caused by ethiologic heterogeniety. 

Regarding the review components, extensive review on geographic-ethnic distribution (paragraph 2) in this paper seems heterotopia. On the other hand, the authors did not touch enough on the genetic basis which is inevitable for the fundamental paper like this. There were some authors that had the same view about ethiologic heterogeniety who used genetic point of view as a tool to address the theory. For example;

  • Sangkhathat S, Laochareonsuk W, Maneechay W, Kayasut K, Chiengkriwate P. Variants Associated with Infantile Cholestatic Syndromes Detected in
    Extrahepatic Biliary Atresia by Whole Exome Studies: A 20-Case Series from
    Thailand. J Pediatr Genet. 2018 Jun;7(2):67-73.
  • Vij M, Rela M. Biliary atresia: pathology, etiology and pathogenesis. Future
    Sci OA. 2020 Mar 17;6(5):FSO466.

In addition, genetic predisposition such as ADD3 polymorphisms, EFEMP1 polymorphisms, etc. that were associated with the disease should be addressed as the frame of the paper was a review on BA etiology.

Table 1: Using the terms 'non-syndromic' as a subcategory of 'syndromic BA' can be confusing. Can the main cateegory be 'BA with associated anomalies'?

Regarding viral theory, although CMV was found in a large proportion of isolated BA, other viruses were reported and should be reviewed in depth. 

Author Response

Reviewer #2

The manuscript is a comprehensive review on pathogenesis of biliary atresia with a focus on the concept of common inflammatory pathway caused by ethiologic heterogeniety.

The aim of the paper was to review all clinically relevant mechanisms that could be supported by a certain level of evidence in the human condition.  The starting point was the clinical framework that I have postulated before, indeed since the publication of  Hartley et al. (Biliary Atresia . Lancet. 2009 Nov 14;374(9702):1704-13).  Four variant classes are defined: Syndromic; Cystic; CMV-associated and Isolated.

Regarding the review components, extensive review on geographic-ethnic distribution (paragraph 2) in this paper seems heterotopia. On the other hand, the authors did not touch enough on the genetic basis which is inevitable for the fundamental paper like this. There were some authors that had the same view about ethiologic heterogeniety who used genetic point of view as a tool to address the theory. For example;

I am not certain what “heterotopia” refers to.   I have increased the genetic elements within the paper, though in the context of isolated BA the size of the  contribution of susceptibility genes is far from clear.

Sangkhathat S, Laochareonsuk W, Maneechay W, Kayasut K, Chiengkriwate P. Variants Associated with Infantile Cholestatic Syndromes Detected in

Extrahepatic Biliary Atresia by Whole Exome Studies: A 20-Case Series from Thailand. J Pediatr Genet. 2018 Jun;7(2):67-73.

This is an interesting paper but I would argue that in many of these cases they are examples of the diseases listed with enough biliary hypoplasia to mimic BA and diagnose “biliary atresia”. Not all infants that have had a Kasai operation are BA. Alagille’s syndrome and PFIC are especially suspect in this regard.  I have operated on two infants with proven Mitchel Riley syndrome, both with biliary pathology, one with missing parts of the bile duct i.e. “biliary atresia”, the other was intact but with marked conjugated jaundice. These were both completely different to the pathology found in any of my biliary atresia series – non-inflammatory, with unusual duodenal pathology and pancreatic failure.

Vij M, Rela M. Biliary atresia: pathology, etiology and pathogenesis. Future

Sci OA. 2020 Mar 17;6(5):FSO466.

This is a review, mentioning genetics certainly but does not give it as the primary cause.

In addition, genetic predisposition such as ADD3 polymorphisms, EFEMP1 polymorphisms, etc. that were associated with the disease should be addressed as the frame of the paper was a review on BA etiology.

New Section added

Table 1: Using the terms 'non-syndromic' as a subcategory of 'syndromic BA' can be confusing. Can the main cateegory be 'BA with associated anomalies'?

This is rather unconventional, I agree, but actually conveys the position…. There is an association with other anomalies (e.g. oesophageal atresia) which defies categorisation as “syndromic”.  

Regarding viral theory, although CMV was found in a large proportion of isolated BA, other viruses were reported and should be reviewed in depth.

Section expanded, with timeline of ebb and flow for reovirus.  This will be taken further in the Viral chapter, of which I am a co-author.